# Conjoint Analysis: A Research Method to Study Patients’ Preferences and Personalize Care

**DOI:** 10.3390/jpm12020274

**Published:** 2022-02-13

**Authors:** Basem Al-Omari, Joviana Farhat, Mai Ershaid

**Affiliations:** 1Department of Epidemiology and Population Health, College of Medicine and Health Sciences, Khalifa University, Abu Dhabi P.O. Box 127788, United Arab Emirates; joviana.farhat@ku.ac.ae (J.F.); mai.ershaid@ku.ac.ae (M.E.); 2KU Research and Data Intelligence Support Center (RDISC) AW 8474000331, Khalifa University of Science and Technology, Abu Dhabi P.O. Box 127788, United Arab Emirates

**Keywords:** conjoint analysis, multivariate technique, rating, ranking, choice based, patient preferences

## Abstract

This article aims to describe the conjoint analysis (CA) method and its application in healthcare settings, and to provide researchers with a brief guide to conduct a conjoint study. CA is a method for eliciting patients’ preferences that offers choices similar to those in the real world and allows researchers to quantify these preferences. To identify literature related to conjoint analysis, a comprehensive search of PubMed (MEDLINE), EMBASE, Web of Science, and Google Scholar was conducted without language or date restrictions. To identify the trend of publications and citations in conjoint analysis, an online search of all databases indexed in the Web of Science Core Collection was conducted on the 8th of December 2021 without time restriction. Searching key terms covered a wide range of synonyms related to conjoint analysis. The search field was limited to the title, and no language or date limitations were applied. The number of published documents related to CA was nearly 900 during the year 2021 and the total number of citations for CA documents was approximately 20,000 citations, which certainly shows that the popularity of CA is increasing, especially in the healthcare sciences services discipline, which is in the top five fields publishing CA documents. However, there are some limitations regarding the appropriate sample size, quality assessment tool, and external validity of CA.

## 1. Introduction

The popularity of conjoint analysis (CA) in health outcomes research has been increasing in recent years [1,2]. Yet, the untraditional concept of this research method is still unclear for many healthcare researchers and clinicians in terms of the design complexity and the absence of confirmed sample size [1,3,4]. Throughout clinical practice, healthcare professionals have been driving their efforts towards a patient-oriented profession to improve patient adherence to medications, prognosis, and quality of life [5]. Over the years, approaches that are referred to as stated and revealed preference methods have been executed to assess patients’ preferences in relation to drug pharmacodynamics, pharmacokinetics, and financial characteristics [5,6]. The stated preferences method relies on what people state while evaluating alternative hypothetical situations (hypothetical decision) [7,8]. Contrarily, the revealed preferences method relies on the observations of the actual choices made by people to measure preferences (actual decision) [7,8,9]. Since stated choice does not always reveal the actual preference [8], behavioral scientists developed alternative techniques that involve studying choice behavior rather than just stated choices [10]. Therefore, the application of alternative methods in clinical practice is linked to the patients’ own perspective of selecting the best cost-effective treatment, while considering their social and psychological situation instead of relying mostly on disease symptoms [6,11,12].

One of the main methods of examining patients’ preferences is the CA, which was developed to scrutinize preferences within the decision-making process [4,13]. CA is a stated preference method that measures how respondents state that they will react in a certain situation [14,15]. Conjoint measurement was first developed in the 1960s by the American mathematical psychologist Duncan Luce and the statistician John Tukey [16]. In the early 1970s, Green and Rao introduced conjoint measurement to marketing research in order to understand and predict buyer behavior [17] and thereafter it was most widely used in marketing research [18]. Although the CA technique was developed in the 1960s, it was not until the 1990s that it was used to elicit patients’ views in the healthcare field [19]. Since then, its popularity and social impact have been growing gradually through its frequent usage in health services rating based research studies [1,3].

The term CA generally belongs to the description of the variety of quantitative methods used to analyze preferences [19,20]. The denomination “conjoint” refers to the idea that several factors can be “considered jointly” [21]. Therefore, CA permits people to choose between different hypothetical products or treatments scenarios rather than evaluating their characteristics separately. CA presents people with ideas that closely resemble the decisions made in real life when choosing between alternatives [22]. For example, if a patient is requested to select the preferred surgical procedure from several alternatives for the treatment of kidney stones, they may consider a specific procedure superior to others. CA elicits patients’ preferences for the selected surgical procedure by evaluating multiple factors associated with each offered procedure. These factors may include adverse events, associated benefits, recovery time, and cost.

When people are making treatment decisions, they base their choice on several characteristics of this treatment. CA assumes that each one of these characteristics has a specific importance to people and they are making trade-offs between these characteristics [4,23]. It is also suggested that people give well-ordered preferences when evaluating options together rather than in isolation [17]. Therefore, unlike traditional questionnaires, CA poses several hypothetical scenarios and asks patients to rate, rank, or choose their preferred scenario [24]. Accordingly, the importance of CA is highlighted by being a multivariate technique used specifically to understand how respondents develop preferences for products or services [25]. In most cases, individuals could make up their minds about a particular treatment characteristic, but they might change their preference when this characteristic is combined to form a treatment scenario. For example, a patient’s decision to choose between different pain-relieving medications could be based on several characteristics of these medications, which may include a pain-relieving effect, frequency of administration, and side effects. This patient may state that she/he would like the medication that provides maximum pain relief, is taken once a day, and has no side effects. Yet, she/he may change their decision when the reality states that the medications with maximum benefits hold risks of side effects and may have to be taken more than once a day. In this situation, the patient must trade off benefits, frequency, and side effects and make some compromises. Hence, CA suggests that presenting patients with several scenarios in a conjoint task could resemble the decision made when selecting medication in real life.

CA methods use three main approaches and tools to elicit well-ordered preferences: ranking, rating, or discrete choices [19]. When conducting a CA questionnaire, researchers can either utilize a pre-developed questionnaire design or develop their own customized one. For example, Ratcliffe and colleagues developed their conjoint questionnaire using a computer software package to produce a fractional factorial design [26]. Others built narratives describing different options and asked participants to rate these options [27] or created hypothetical scenarios and asked participants to choose between them [22]. In recent years, pre-developed designs such as adaptive conjoint analysis (ACA) or adaptive choice-based conjoint (ACBC) by “Sawtooth software” (a provider of CA software packages) are becoming popular [4,28,29]. These designs provide researchers with questionnaire templates and an analysis platform. Consequently, researchers can customize the template to suit their requirements and can build up the questionnaire using the attributes and levels specific to their study. Then, a built-in statistical software such as hierarchical basin (HB) can be used to analyze the data.

## 2. Methods

This article aims to provide an overview of the CA method and analyze the growth of its application over the past 70 years. It also narratively discusses the literature of the CA method’s process and validity, its use in healthcare settings, and its strengths and limitations.

A comprehensive literature search was conducted. Following Gasparyan and colleagues’ recommendations [30], PubMed (MEDLINE), EMBASE, Web of Science, and Google Scholar were electronically searched without language or date restrictions. Keywords related to “conjoint analysis”, “discrete choice”, “choice experiment”, “rating conjoint”, and “ranking conjoint” were used to search the literature. Additionally, the lead author has significant experience in the field, and the opinions expressed in this article are also based on personal experience of writing, editing, and commenting on reviewed articles.

The Web of Science Core Collection (WoSCC) was utilized to identify the trend of publications and citations over the past 70 years. WoSCC is a database providing access to billions of cited references dating back to 1900 in the areas of life sciences, social sciences, arts, and humanities [31], and is an emerging source of citation index [32,33,34,35]. Bibliometric studies, which are used to systematize and summarize the growing body of publications [36] and focus on a topic’s popularity at a given point in time [37], mainly use WoSCC. Therefore, an online search was conducted utilizing all databases indexed in the WoSCC to identify the publications and citations trend in CA. The retrieved database was searched on the 8th of December 2021. The database was accessed through the electronic library portal of Khalifa University, United Arab Emirates. The Boolean search query method was applied. The searching key terms covered a wide range of synonyms which included “conjoint analysis” OR “conjoint measurement” OR “conjoint studies” OR “conjoint choice experiment” OR “discrete choice conjoint experiment” OR “discrete choice experiment” OR “pairwise choices” OR “Best-Worst Scaling” OR “Best Worst Scaling” OR “MaxDiff Scaling” OR “Maximum Difference Scaling” OR “ranking conjoint” OR “rating conjoint” OR “adaptive conjoint analysis” OR “adaptive choice based conjoint” OR “choice based analysis” OR “full profile conjoint” OR “choice based conjoint” OR “choice set” OR “relative preference weight” OR “hypothetical scenario” OR “stated preference”. The search field was limited to the title, and no language or date limitations were applied.

## 3. Conjoint Analysis Trend over the Past 70 Years

The WoSCC search identified a total of 9614 documents related to CA, which were published between 1950 and the 8th of December 2021. The result of the search demonstrated a significant increase in the production and citation of published papers related to CA over the years to reach nearly 900 documents and 20,000 citations in 2020 and 2021 (see Figure 1). The gradual increase in citations and research production indicates the expanded popularity of CA methods. Furthermore, it is an indication of the improvement of the reporting of conjoint experiments that are conducted for commercial purposes. Between 1981 and 1985, it was estimated that approximately 400 commercial conjoint analysis applications were carried out each year [38]. Yet, only a few documents were published each year during the 1980s. This indicates that the primary purpose of using CA during that period was commercial and academic application and reporting have only become popular during the last 20 years. Furthermore, many advances in CA methods were documented during the 1980s and 1990s [39] along with observations of greater interest in CA usage throughout the healthcare field during the 1990s [19].

The results of the citations analysis indicated that the business and economics field has the highest number of publications of CA. This is expected, as CA originated from this area of research, more specifically for marketing research. It is not surprising that the healthcare field of research was one of the top five areas publishing papers on CA topics. This indicates the growing interest in the CA method by healthcare researchers (see Table 1). In terms of the type of documents, the highest number of published documents were research articles (n = 7047; 73.3%), then meeting abstracts (n = 1624; 16.9%). Furthermore, the highest contributing countries to CA research were the USA (n = 2321; 24.1%), People’s Republic of China (n = 1549; 16.1%), and England (n = 899; 9.4%).

## 4. The Conjoint Analysis Study Process

During the 1990s, the initial focus of researchers was to assess patient preferences and satisfaction regarding the treatment outcome only [25]. This was evidenced by the large number of health studies assessing patients’ quality-adjusted life years (QALYs) and healthy-years equivalents [24]. By the year 2000, the use of preferences methods gradually increased in the healthcare setting. Ryan and Farrar aimed to familiarize and engage patients with their treatment plan in cooperation with their physicians by allowing them to exhibit their preferences. This not only considered patients’ treatment response but also treatment characteristics, surgical options, as well as physicians’ care and attitude [16]. Ryan and Farrar stratified a multistep plan in order to practice a standardized CA study and achieve a precise assessment of patients’ preferred choices through five main stages stated as follows.

### 4.1. Identifying the Relevant Attributes

Attributes are known to be the factors, features, or characteristics which are believed to influence people’s preferences for a particular product or treatment [40,41]. Identifying attributes must be supported by evidence that suggests the potential range of preferences and values that people may hold [42]. Attributes must also be balanced between what is important to the respondent and what is relevant to decision-makers [42]. Selecting and defining the attributes can be achieved through reviewing the literature, healthcare experts’ group discussions, and interviewing individual subjects from the patients and public involvement (PPI) groups [41]. In some cases, a predefined policy question may be already available. Defining attributes is the most fundamental and critical aspect of designing a good CA study [43]. Therefore, attributes must be written in terminology that is easy for patients to understand. This was achieved when the wording and terminology of the attributes were based on the research users’ group (RUG) recommendations and suggestions [23]. Table 2 shows examples of different attributes for pain-relieving medication.

### 4.2. Assigning Levels

In CA, each attribute is defined by a series of levels [28]. Therefore, assigning levels to attributes will follow up from identifying attributes and is considered important. Levels represent the different alternatives for each attribute and must be reasonable and capable of being traded off against each other [44]. Bridges and colleagues suggested that researchers should avoid the use of ranges to define attributes (such as a copayment from USD 5–10) because this requires the respondent to subjectively interpret the levels, which will affect the results, and they should also be cautious of choosing too many levels [42]. Furthermore, levels of unrealistic and extreme values should be avoided as they will not be acceptable to respondents [23]. Table 2 shows examples of levels for pain-relieving medication attributes.

### 4.3. Choosing Scenarios

Once the attributes and levels are identified, the levels of all attributes are combined to form all possible scenarios. CA tasks are the mechanism by which possible profiles are presented to respondents for preference elicitation [42]. The higher the number of attributes and levels, the higher the number of possible scenarios. Therefore, researchers can very rarely use all produced scenarios. Instead, CA studies utilize the orthogonal fractional factorial experimental designs to construct a set of hypothetical scenarios [45]. If the scenarios are described with respect to all of the attributes being studied, this is referred to as a full-profile choice experiment [46]. For example, if the treatment studied has six important attributes, a full-profile scenario would describe the treatment of all six attributes. This means that each scenario would have all six levels; one from each attribute. When a study includes a large number of attributes, it becomes complex for participants to process these scenarios. Instead, the concept of partial-profile choice experiments has been proposed to estimate preferences for a large set of attributes [47]. In the partial-profile choice experiment, each scenario includes a subset of the total number of attributes being studied [18,48]. All attributes are randomly rotated into the different scenarios, so across all scenarios in the experiment, each respondent typically considers all attributes and levels [18,48,49]. For example, if the treatment to be studied has 12 attributes, a partial profile choice experiment describes the treatment on a few attributes in each scenario, i.e., each scenario would possibly have six characteristics of the treatment. The main issue with the partial-profile choice experiment is that the data are spread quite thinly because each task has many attribute omissions [50,51]. This task assumes that respondents can ignore omitted attributes and base their choice solely on the partial information presented in each task [51].

### 4.4. Establishing Preference

People’s preferences of the developed scenarios can be established using a rating, ranking, or choice-based approach. The ranking approach in Table 3 asks respondents to list the scenarios in order of preference [19], whereas the rating approach in Table 4 requests respondents to assign a score to each scenario, e.g., 0–10 or 0–100, usually presented on a visual analog scale [52]. Accordingly, the rating/ranking CA approach usually requires a low cognitive load and can be easily implemented, but they are direct scaling methods that impose the lack of clear trade-off between preferences [53]. The choice-based approach in Table 5 asks respondents to choose their preferred scenario out of a couple or few scenarios [54]. The choice-based approach presents multiple attributes susceptible to simultaneous assessment [55], then allows patients to pick the best available option from a set of scenarios which in turn permits patients to make clear trade-offs between different levels [56]. However, with the choice-based approach, there is a lack of ability to justify the reasons behind the choices made [57]. Rating, ranking, and choice-based CA approaches can be presented to respondents in various ways using different techniques. So, these conjoint methods could be designed through hard copies of pen and paper questionnaires or via computerized software programs.

In Table 5, each vertical column represents a group of characteristics specific to medication A, B, and C, while each row illustrates the levels of a particular attribute in each of the medication scenarios. The participants are asked to select their preferred scenario and therefore, they have to trade-off attributes and levels against each other.

### 4.5. Analyzing Data

In CA methods, regression techniques are used to estimate the relative importance of the attributes and the utilities (part-worths) of the levels [19,23]. Moreover, the maximum amount of money that the patients are willing to pay for service or treatment is known as the willingness to pay (WTP) [44,58]. The part-worths are interval data within each attribute that represent the utilities of the levels within that attribute. Generally, part-worths are scaled to an arbitrary additive, while the relative importance is percentages data that are given to each attribute. The higher the percentage the more important is that attribute to the respondents and the relative importance of all attributes add up to 100%.

## 5. Conjoint Analysis in Healthcare

Several traditional quantitative and qualitative methods have been used to study patients’ preferences. However, these studies were limited in number [59]. These methods have been used to accentuate patients’ preference in general but did not identify the trade-off that patients make to one of the treatment factors against another [60]. Unlike traditional questionnaires, the CA method can be used to study preferences [13] and quantify the trade-off that patients do between the different treatment factors [61].

Nowadays, there has been a rapid increase in the use of CA to quantify preferences for various healthcare services and treatment options [62]. For example, in clinics where the majority of cases are non-urgent, CA was found to be a useful tool for allowing patients to discuss their needs and choose medication, health service, and diagnostic tests that suit them the most [63,64]. In turn, CA constituted a supportive tool for clinicians to better understand patients’ preferences and individualize their treatment plans [24,65,66]. Hence, the CA use in practice was suggested as a tool to effectively strengthen the communication between patients and healthcare professionals and to engage both parties in the shared decision-making process [12,62]. A scoping review of the studies using CA recommends the use of CA to identify patient preferences for mental health services, which could improve the quality of care and increase the acceptability and uptake of services among patients [67].

In the hospital setting, implementation of CA constituted a feasible and useful tool in several clinical areas such as investigating hospital stakeholders’ decision-making in the adoption of evidence-based interventions [68], eliciting patients preferences regarding diagnostics and screening [69,70], improving decision-making regarding patients’ treatment [71,72], understanding patients’ perceived needs and expectations [73], and determining the clinical factors that physicians prioritize regarding patients’ treatment [74]. Furthermore, the CA method was very useful during the recent pandemic, as it has been used to understand how people prioritize when deciding whether to present to the emergency department during the coronavirus disease (COVID-19) pandemic for care unrelated to COVID-19 [75]. This understanding of patients’ priorities helps healthcare professionals to structure an appropriate patient safety assessment which led to remarkably reduced chances of deaths, departments’ crowdedness, and spread of infections [75]. Moreover, CA was seen to efficiently assess the cost-effectiveness value of alternatives when patients are requested to select their preferred treatment option taking into consideration their financial situation [76]. Thus, CA is a practical method for generating treatments’ marketing decisions in the pharmaceutical industry, which highly relies on patients’ preferences about a specific drug.

Throughout the discussed CA-based studies, we can discern that CA is a major asset for valuing patients’ important contributions in the decision-making process by assessing patients’ preferences in accordance with treatment selection, patient care, offered health services, and cost comparisons. Research-based literature verified that CA can be a practical method of estimating utility for any combination of attributes, including combinations representing goods or services which may not currently be available [77]. Thus, when studying patients’ preferences for treatment, CA allows patients to choose their preferred treatment option based on the treatments’ characteristics in isolation of treatments’ names or market brands. Therefore, the precise performance of CA can offer valuable approximation in relation to the relative importance of different aspects of care, the trade-offs between these aspects, as well as the total satisfaction or utility that respondents obtain from healthcare services [19].

## 6. Validity of Conjoint Analysis Data

Recently, it has been suggested that the value of CA is not only related to its elevated frequency of implementation but also to the unique ability of this method to generate individuals’ preferences realistically enough to match various decision-making processes faced by individuals in the real world [78]. Assessing the validity of CA means examining the ability of this data collection tool to accurately measure what it is supposed to measure [79,80]. There are many types of validity and some of these types may overlap, and researchers may argue the names of different types of validity; for example, face validity is often confused with content validity [81]. Over 20 years ago, very little was known whether CA works in predicting significant real-world actions [82]. However, within the last couple of decades, researchers have been studying and investigating the validity of CA. Despite the high expansion of CA usage in market and healthcare research, CA validity studies are still limited. A large validity study including over 2000 commercial CA research indicated that there was no validity gain for CA over time [83]. This could be due to the variation in the CA tools, the expansion of recruitment methods to online and social media, and the differences in the estimation parameters used for each study. In general, the validity for all CA tools can be measured in two ways:External validity is the ability of the CA tool to predict what people would choose in real life. This can be achieved by asking the question “did people choose what CA predicted?”. For example, in a conjoint study estimating the market share for an American multinational telecommunications corporation, various trial simulations were implemented hypothesizing that several product features had to be changed in order to attain desired sales (8% of the total market share) [56]. Four years after launching this product, the actual share was just under 8% [56], concluding that CA contributes towards the identification of people-desired choices and the estimation of the actual preference behavior. Investigating external validity for CA methods is a challenging task that requires the researcher to follow the participants to examine if they did what the CA tool predicted in terms of buying a product, taking a treatment, attending a particular doctor’s clinic, etc.Internal consistency validity is the main validity criterion that has been studied in recent years for strengthening the reliability and applicability of CA. To test the internal validity, the holdouts’ choices are used [84]. The holdouts are choices that are similar to those selected by the participants in real life but are “held out” of the conjoint approximation by not being part of the final estimation. The internal validity of the conjoint task is examined by comparing how well conjoint utilities predict choices from the holdout tasks. Therefore, the holdout tasks are not used in the estimation of part-worths, but they are presumed to represent respondent choices in the real world [85]. In a review evaluating CA as a method of estimating consumers’ preferences, Green and Srinivasan reported that several studies have demonstrated the consistency of conjoint models in terms of reproducing current market conditions [39]. Furthermore, a study offering four topical antibiotics to treat acne confirmed CA consistency and validity when patients’ preferences assessment, the simulated product rankings, and the results of the traditional questionnaire were matched [86].

Generally, as the use of CA in the healthcare setting increased, some validity studies were performed to approach more patients’ preferences and expectations. These were not focused only on the patients’ benefit–risk trade-offs, but also on evaluating the patients’ WTP for treatments or services [55].

## 7. Strengths and Limitations of Conjoint Analysis

Over the years, CA design enabled researchers to elicit and quantify patients’ preferences for treatments and services using a smaller number of scenarios extracted from a larger pool of choices [1,19,72,87,88,89]. CA is well known for providing easy experimentation for aspects such as price and features before launching a new product, treatment, or health service [90]. It is suggested that when people are deciding between multi-attributes alternatives, they apply an unconscious scoring mechanism or system of their preference weight; CA is capable of uncovering this system [18]. This is achieved by providing respondents with the opportunity to make trade-offs between the specific features of competing items to reach final realistic decisions [18,91]. These trade-offs are based on the value that people place on each attribute.

Some of the limitations of the CA methods are due to the lack of validated quality assessment tools for CA studies and lack of consensus on appropriate sample sizes [1,3,4]. Furthermore, one of the inherent limitations of CA is that respondents are evaluating hypothetical scenarios, which might be different from what they do in real life [59]. It is suggested that the CA questionnaire fatigues respondents as it takes more time to complete than traditional questionnaires and it requires more focus and concentration [92]. It is also suggested that many patients are not well exposed to research surveys [57,93]. Therefore, reconsidering the number of questions and alternatives presented to participants during data collection is vital to avoid unnecessary respondents’ fatigue [1,4].

## 8. Strengths and Limitations of This Study

To the best of our knowledge, this is the first article to report the trend of CA publications and citations over the past 70 years and the increase in its popularity based on the amount of published literature. This article takes a well-defined, rich, and clear approach to the discussion of CA. It provides a summary of the very large and wealthy literature describing CA methods. The narrative nature of this article is based on a comprehensive search of the literature and utilized several databases. However, the narrative nature of the discussion could be subjective and open to different interpretations. Therefore, we recommend that the results of this article must be interpreted in line with its limitations. The results in relation to the CA trend over the past 70 years are not based on a comprehensive bibliometric analysis in terms of visualization. Nonetheless, it is based on a comprehensive search of WoSCC databases to identify the growth in CA publications.

## 9. Conclusions

The popularity of CA in healthcare has been increasing and its use in this setting is gradually competing with its use in business and marketing research. CA is a useful method for eliciting patients’ preferences and WTP. However, there are some limitations in the available CA literature, specifically regarding the appropriate sample size, quality assessment tool, and the validity of CA. This highlights the need for researchers from different fields that use CA methods to come together and develop tools to address these limitations.

## Figures and Tables

**Figure 1 jpm-12-00274-f001:**
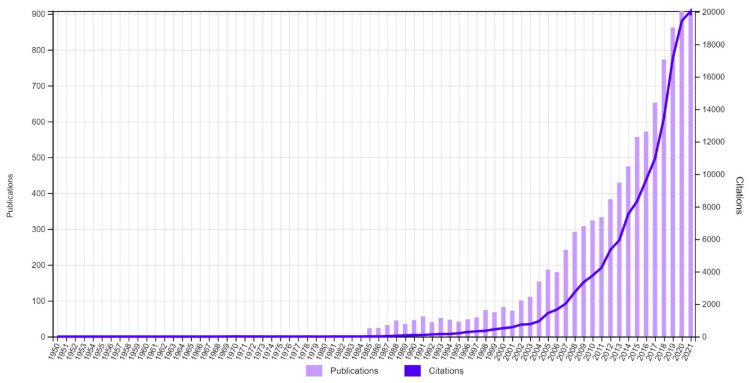
The trend of CA documents published between 1950 and 2021.

**Table 1 jpm-12-00274-t001:** Top 10 research areas publishing CA documents.

Research Areas	Number of Published Papers
Business Economics	n = 3663
Computer Science	n = 2652
Mathematics	n = 2495
Engineering	n = 2397
Healthcare Sciences Services	n = 1729
Psychology	n = 1624
Behavioral Sciences	n = 1365
Environmental Sciences Ecology	n = 1145
Science Technology Other Topics	n = 787
Public Environmental Occupational Health	n = 629

Note: The number of published papers in Table 1 adds up to more than the total analyzed documents (n = 9614). The reason for this is that several documents are classified by the databases under several research areas; for example, some documents would be classified under psychology and healthcare sciences services at the same time.

**Table 2 jpm-12-00274-t002:** An example of attributes and levels for pain-relieving medication.

Attributes	Levels
Frequency of administration	Once a day Twice a day Three times a day When needed
Type of medication	Prescription drug Non-prescription drug
Route of Administration	Orally Injection Topical
Therapeutic effect	Relief of mild pain Relief of moderate pain Relief of severe pain
Adverse events	Low risk of stomach pain Moderate risk of stomach pain High-risk stomach pain
Insurance cost coverage	Completely covered by the insurance Partially covered by the insurance Not covered by the insurance

**Table 3 jpm-12-00274-t003:** Examples of the ranking approach.

Each Column Represents a Medication. Please Rank These Medications from the MOST Preferred (1) to the LEAST Preferred (3).
Attributes	Medication “A”	Medication “B”	Medication “C”
Frequency of administration	Three times a day	Once a day	When needed
Type of medication	Prescription drug	Non-prescription drug	Prescription drug
Route of administration	Topical	Oral	Injection
Therapeutic effect	Relief of severe pain	Relief of moderate pain	Relief of moderate pain
Adverse events	High-risk stomach pain	Moderate risk stomach pain	High-risk stomach pain
Insurance cost coverage	Covered by the insurance	Not covered by the insurance	Partially covered by the insurance
Rank	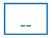	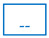	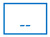

**Table 4 jpm-12-00274-t004:** Examples of the rating approach.

How Likely Are You to Take the Medication Below?Slide the Pointer to the Position on the Scale to Indicate Your Answer. A “0” Means You Definitely Would NOT Take This Drug and a “50” Means You Definitely Would Take This Drug.
Once a dayNon-prescription drug OralRelief of moderate painModerate risk stomach painNot covered by the insurance
Definitely would NOT take		Definitely would take
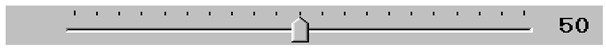

**Table 5 jpm-12-00274-t005:** Examples of the choice-based approach.

Each Column Represents a Medication. Please Select the ONE Medication That You Prefer the Most.
Attributes	Medication “A”	Medication “B”	Medication “C”
Frequency of administration	Three times a day	Once a day	When needed
Type of medication	Prescription drug	Non-prescription drug	Prescription drug
Route of administration	Topical	Oral	Injection
Therapeutic effect	Relief of severe pain	Relief of moderate pain	Relief of moderate pain
Adverse events	High-risk stomach pain	Moderate-risk stomach pain	High-risk stomach pain
Insurance cost coverage	Covered by the insurance	Not covered by the insurance	Partially covered by the insurance
	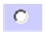	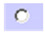	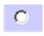

## Data Availability

Not applicable.

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
