# Peer review of "Conjoint Analysis: A Research Method to Study Patients’ Preferences and Personalize Care"

_jpm, 2022, doi:10.3390/jpm12020274_

Round 1

Reviewer 1 Report

Dear authors,

Congratulations on the manuscript. It's an interesting topic, with relevance to the current healthcare trends, which are mainly approaching patient-centered interventions and models, rather than biomedical phenomena.

I have some suggestions and comments. Also, the manuscript would benefit from a moderate English revision, to avoid some typos.

ABSTRACT

  • There's nothing related to the method used. A line or two would be appropriate.

2. Conjoint analysis trend over the past 70 years

  • Check the first paragraph of this section. It seems part of the journal's template.
  • This section is describing the methods you used. As a general suggestion, I think you should revise it and present a more robust method section, as it is very poor. Even though your manuscript is of a narrative and descriptive nature, research always requires a robust method description.
  • You should enumerate all databases searched, and then mention that WoSCC was used as a search engine.  As such: why did you only use WoSCC and not other search engines? Do you think this is a limitation of your study? You might've lost some important results. In this sense, also move the paragraph after Table 1 to a dedicated section about Limitations, at the end of your manuscript.
  • It's not clear what type of literature review are you using in your study. Maybe a narrative review or a literature review. Could you clarify?
  • Did you find any previous studies similar to yours? It would be important to know, so you could justify why this study is important. If there are previous studies, why do you think it's important to perform this new analysis of the topic?
  • You found 9.614 documents. Then, in the paragraph after Figure 1, you have the type of documents, which in total sum 13.440. Did you analyze all 9.614 documents? Did you exclude some? What were your inclusion and exclusion criteria? This is part of a more robust method section that you should perform. If I wanted to repeat your review, as it is, I couldn't understand how you conducted it.
  • How do you explain the exponential increase in the citations and research production about CA until the year 2021? Also, as you said in the Introduction, in the 60's the technique was first developed, and in the 70's is was applied to marketing. There are no documents on those years, when we analyze your Figure. How do you explain that? So, why only in 1985 do we start to see documents and citations? Might this be a limitation of your study?
  • Specify what's Peoples R China.
  • Don't include the year 2022 in Figure 1, as obviously it wasn't included in your timeframe.

3. Conjoint analysis study process

  • It's not clear what kind of qualitative/quantitative instruments or tools are used in CA. You've presented self-reported instruments. Are there other types of tools? Are they always the same throughout CA application? Does the researcher choose the one he finds more adequate? or should he follow a pre-developed structure? Could you clarify this earlier in the paper?

4. Conjoint analysis in healthcare

  • Your paper should be a way to convince researchers and clinicians to use CA over other statistical methods. As such, you should include some paragraphs with a description of why this method seems more adequate than other similar methods.

5. Validity of conjoint analysis data

  • If there are several different surveys or forms that the researcher can use within CA, how do you perform a validity test and use a validated version in a study? This relates to the concern I've raised in section 3.

Limitations of Your Study

  • You should include a paragraph about the limitations of your study. I can find some: the review is only based on one database, you've probably lost some other studies and documents relevant to understanding CA.
  • Do you consider this a limitation: using a statistical technique developed and used in marketing research in healthcare sector?

Conclusions

  • The conclusion should be in line with the aims, which you don't mention in this section.

References

  • 33% (n=24) of your references have more than 10 years. If possible, update some.
  • References 22, 56 and 57 have no year of publication, please revise.

Author Response

On behalf of the authors, I would like to thank you for allowing us the opportunity to revise our manuscript. The authors would also like to thank you for the comments and recommendations that we believe have improved the quality of the manuscript. The authors have made amendments to address all comments. We have made sure that we responded to every comment and considered all possible amendments to the manuscript. Please see below point-by-point responses and find attached our manuscript “Conjoint analysis: a research method to study patients’ preferences and personalize care.” –Manuscript ID: jpm-1545811 for your consideration.

Kind regards

Basem Al-Omari

Congratulations on the manuscript. It's an interesting topic, with relevance to the current healthcare trends, which are mainly approaching patient-centered interventions and models, rather than biomedical phenomena.

I have some suggestions and comments. Also, the manuscript would benefit from a moderate English revision, to avoid some typos.

The authors would like to thank the reviewer for allowing us the opportunity to revise our manuscript. The authors also would like to thank the reviewer for taking the time to comprehensively review our manuscript and providing comments and recommendations that we believe have improved the quality of the manuscript. All amendments to the manuscript are made in blue font to facilitate your review. The authors summarised the responses in several rows for clarity and to make sure that we do not miss any of the reviewer’s comments. As per your recommendation, the manuscript went under English revision.

ABSTRACT

  • There's nothing related to the method used. A line or two would be appropriate.

As recommended, the authors wrote a short methods section in the abstract (lines 15-21).

2. Conjoint analysis trend over the past 70 years

  • Check the first paragraph of this section. It seems part of the journal's template.

The authors agree with the reviewer. This paragraph was removed.

  • This section is describing the methods you used. As a general suggestion, I think you should revise it and present a more robust method section, as it is very poor. Even though your manuscript is of a narrative and descriptive nature, research always requires a robust method description.

Thank you for this important comment. The authors added a new section describing in detail the methods used to search the literature and the methods used to identify the trend in CA publications (lines 105-136).

  • You should enumerate all databases searched, and then mention that WoSCC was used as a search engine.  As such: why did you only use WoSCC and not other search engines? Do you think this is a limitation of your study? You might've lost some important results. In this sense, also move the paragraph after Table 1 to a dedicated section about Limitations, at the end of your manuscript.

The authors would like to thank the reviewer for raising these questions. To clarify the use of WoSCC as the search database to identify the publication trend, the authors added justification of this with supporting evidence in the new method’s section (lines 118-125). WoSCC has been used as the citation index to identify the trend of publication in certain areas of research in the majority of bibliometric analyses. Therefore, the authors believe that its use in this context is appropriate. However, the search for the CA literature used in the manuscript utilised PubMed (MEDLINE), EMBASE, and Web of Science. This has now been added to the methods section (lines 110-117).
As recommended, the authors added a strengths and limitations section before the conclusion and moved the paragraph after table 1 to this section (lines 387-399).

  • It's not clear what type of literature review are you using in your study. Maybe a narrative review or a literature review. Could you clarify?

The authors clarified this by adding a short paragraph at the beginning of the methods section describing the aim and nature of the article (lines 106-109).

  • Did you find any previous studies similar to yours? It would be important to know, so you could justify why this study is important. If there are previous studies, why do you think it's important to perform this new analysis of the topic?

Some studies discuss a specific type of CA method. However, to the best of our knowledge, this is the first study to take this approach to discuss CA and provide numbers of the published literature to support the claim that CA popularity in healthcare setting is increasing. The authors added a short discussion of the strengths and importance of this article in the strengths and limitations section (lines 393-399).

  • You found 9.614 documents. Then, in the paragraph after Figure 1, you have the type of documents, which in total sum 13.440. Did you analyze all 9.614 documents? Did you exclude some? What were your inclusion and exclusion criteria? This is part of a more robust method section that you should perform. If I wanted to repeat your review, as it is, I couldn't understand how you conducted it.

The types of documents presented in (lines 162) states: “articles (n=7,047, 73.3%), then meeting abstracts (n=1,624, 16.9%)”. This does not add up to 13,440 but it adds up to 8,671. However, the numbers in table 1 add up to more than the analysed document (n = 9,614), this is because many documents are classified by the databases under several research areas; for example, some documents would be classified under psychology and health care sciences services at the same time. To clarify this, the authors added a note under table 1 describing the reason that the numbers add up to more than the total analysed documents number (lines 167-170).
This section is not a review it is taking a bibliometric analysis approach to identify the trend of the CA published documents from the sources of citation index.

Therefore, it does not require inclusion/exclusion criteria such as systematic reviews. The authors added an explanation of this in the methods section (lines 118-125).

  • How do you explain the exponential increase in the citations and research production about CA until the year 2021? Also, as you said in the Introduction, in the 60's the technique was first developed, and in the 70's is was applied to marketing. There are no documents on those years, when we analyze your Figure. How do you explain that? So, why only in 1985 do we start to see documents and citations? Might this be a limitation of your study?

Many thanks for raising this important point. The authors added an explanation in section 3 (lines 142-151).
Regarding figure 1, there are 0-2 publications per year from 1966 up to 1984. These are not going to be clear in the figure as the purpose of the figure is to show the trend and not the exact number of publications each year. The authors added an explanation of the limited number of studies during this period (lines 144-151).

  • Specify what's Peoples R China.

As recommended by the reviewer, the authors changed the term to: “People's Republic of China” (lines 163).

  • Don't include the year 2022 in Figure 1, as obviously it wasn't included in your timeframe.

As recommended by the reviewer, the authors removed the year 2022 from figure 1.

3. Conjoint analysis study process

  • It's not clear what kind of qualitative/quantitative instruments or tools are used in CA. You've presented self-reported instruments. Are there other types of tools? Are they always the same throughout CA application? Does the researcher choose the one he finds more adequate? or should he follow a pre-developed structure? Could you clarify this earlier in the paper?

It is mentioned in the introduction (lines 60-61) that CA is a quantitative method. Also (lines 75-77): “unlike traditional questionnaires, CA poses several hypothetical scenarios and ask patients to rate, rank, or choose their preferred scenario [22]”.

The full details about the different tools are provided in this section under (4.4. Establishing preference) – (lines 235-271).
The authors would like to thank the reviewer for highlighting that this might not be clear in the manuscript. Therefore, based on the reviewer’s recommendations, the authors provided details to clarify this point at the end of the introduction section (lines 91-104).

4. Conjoint analysis in healthcare

  • Your paper should be a way to convince researchers and clinicians to use CA over other statistical methods. As such, you should include some paragraphs with a description of why this method seems more adequate than other similar methods.

Many thanks for raising this important point. The authors have written the manuscript to deliver this main and most important message. Based on the reviewer’s recommendation, the authors added a paragraph at the beginning of section 5 (conjoint analysis in healthcare) to address this point (lines 273-278).

5. Validity of conjoint analysis data

  • If there are several different surveys or forms that the researcher can use within CA, how do you perform a validity test and use a validated version in a study? This relates to the concern I've raised in section 3.

The authors would like to thank the reviewer for raising this important point. The idea of this section is to discuss the basic concept of CA validity, how it’s measured in general, and to highlight the limited number of studies investigating this issue. There are too many tools that can be used and developed to create CA questionnaires (the authors clarified this in section 3 based on the reviewer’s recommendations). Thus, the discussion in this section is focused on the ways of measuring the validity of CA (internal and external). To clarify this, the authors amended several parts of this section (lines 320-365).

Conclusions

  • The conclusion should be in line with the aims, which you don't mention in this section.

The aim of the paper is clarified in the new methods section (lines 106-109) and in the abstract.

References

  • 33% (n=24) of your references have more than 10 years. If possible, update some.
  • References 22, 56 and 57 have no year of publication, please revise.

Many thanks for raising this point. It is expected that several references are older than 10 years because there is a discussion about the history of CA which was originated in the 1960s. However, there are many contemporary references to support the general and contemporary concepts of the paper.
In response to the reviewer’s recommendations, references 22, 56, and 57 were updated and several new recent references were added to support the discussion in the manuscript.

Reviewer 2 Report

General comment: The authors have tried to compose their work in a better way, but while reading some errors in English are noticed and some paragraphs/sentences are incomprehensible.

Critical comments:

- The referee suggests to categorize the present paper “Review” and not “Article”, because in the aim the authors declare: “this review aims”…

- From the title you expect a lot, but isn’t real research, but a review or an “opinion”. May be, it would be better to reformulate the title.

- Introduction: the aim (lines 29-30), is better to be moved at the end of the section, may be after the line 88.

- Lines 35-45; 62-66: here the authors have used long sentences and aren’t very clear in their meanings.

Note: the word “the patient or this patient” is repeated so often next to each other in sentences…

- Lines 90-96: this paragraph isn’t very clear, especially “the publication of your manuscript implicates that you must make all materials…”.

-Line 112: put in past tense “demonstrates”.

Author Response

On behalf of the authors, I would like to thank you for allowing us the opportunity to revise our manuscript. The authors would also like to thank you for the comments and recommendations that we believe have improved the quality of the manuscript. The authors have made amendments to address all comments. We have made sure that we responded to every comment and considered all possible amendments to the manuscript. Please see below point-by-point responses and find attached our manuscript “Conjoint analysis: a research method to study patients’ preferences and personalize care.” –Manuscript ID: jpm-1545811 for your consideration.

Kind regards

Basem Al-Omari

General comment: The authors have tried to compose their work in a better way, but while reading some errors in English are noticed and some paragraphs/sentences are incomprehensible.

The authors would like to thank the reviewer for allowing us the opportunity to revise our manuscript. The authors also would like to thank the reviewer for taking the time to comprehensively review our manuscript and providing comments and recommendations that we believe have improved the quality of the manuscript. All amendments to the manuscript are made in blue font to facilitate your review. The authors summarised the responses in several rows for clarity and to make sure that we do not miss any response.

As per the reviewer’s recommendation, the manuscript went under English revision.

Critical comments:

The referee suggests to categorize the present paper “Review” and not “Article”, because in the aim the authors declare: “this review aims”…

Many thanks for highlighting this point. The authors believe that this is an article because it includes a review of the literature, an analysis of the publication trend over the past 70 years, an expert opinion, and generated examples to clarify the concept of CA. The authors amended the manuscript, removed the term “review”, added a methods section, and made a significant amendment to the manuscript based on the recommendations of the reviewers.

From the title you expect a lot, but isn’t real research, but a review or an “opinion”. May be, it would be better to reformulate the title.

Please see the response to the above comment. The authors hope that the amendments made to the manuscript would address the expectations from the title. 

- Introduction: the aim (lines 29-30), is better to be moved at the end of the section, may be after the line 88.

The authors would like to thank the reviewer for this important comment. A new methods section has been added and the aim was moved to the beginning of the methods section (lines 106-109).

Lines 35-45; 62-66: here the authors have used long sentences and aren’t very clear in their meanings.

Many thanks for picking up these errors. The authors amended and clarified these sentences (lines 39-44 and 64-70).

Note: the word “the patient or this patient” is repeated so often next to each other in sentences…

- Lines 90-96: this paragraph isn’t very clear, especially “the publication of your manuscript implicates that you must make all materials…”.

-Line 112: put in past tense “demonstrates”

The words “the patient or this patient” have been revised whenever they are next to each other, and sentences were amended to address this comment. 

The paragraph in (lines 90-96) is from the journal template and was in the manuscript by mistake. The authors have removed it.

The term “demonstrates” has been put in the past tense.

Many thanks for highlighting these errors. 

  •  

Round 2

Reviewer 1 Report

Dear authors,

Congratulations, again, on your manuscript and thank you for addressing the suggestions and comments. I hope they were helpful for the discussion and to increase your manuscript's quality.

Good work